# Searching for Effective Methods of Diagnosing Nervous System Lesions in Patients with Alström and Bardet–Biedl Syndromes

**DOI:** 10.3390/genes14091784

**Published:** 2023-09-10

**Authors:** Arleta Waszczykowska, Krzysztof Jeziorny, Dobromiła Barańska, Katarzyna Matera, Aleksandra Pyziak-Skupien, Michał Ciborowski, Agnieszka Zmysłowska

**Affiliations:** 1Department of Ophthalmology, Medical University of Lodz, 90-419 Lodz, Poland; 2Department of Endocrinology and Metabolic Diseases, Polish Mother’s Memorial Hospital Research Institute (PMMHRI), 93-338 Lodz, Poland; jeziorn@wp.pl; 3Department of Pediatric Endocrinology, Medical University of Lodz, 90-419 Lodz, Poland; 4Department of Diagnostic Imaging, Polish Mother’s Memorial Hospital Research Institute (PMMHRI), 93-338 Lodz, Poland; dobaranska@gmail.com (D.B.); katarzyna.matera@iczmp.edu.pl (K.M.); 5Department of Children’s Diabetology, Silesian Medical University in Katowice, 40-752 Katowice, Poland; aleksandra.ewa.pyziak@gmail.com; 6Clinical Research Centre, Medical University of Bialystok, 15-276 Bialystok, Poland; michal.ciborowski@umb.edu.pl; 7Department of Clinical Genetics, Medical University of Lodz, 90-419 Lodz, Poland; agnieszka.zmyslowska@umed.lodz.pl

**Keywords:** Alström syndrome, Bardet–Biedl syndrome, OCT, MRS

## Abstract

Bardet–Biedl syndrome (BBS) and Alström syndrome (ALMS) are rare multisystem diseases with an autosomal recessive mode of inheritance and genetic heterogeneity, characterized by visual impairment, hearing impairment, cardiomyopathy, childhood obesity, and insulin resistance. The purpose of our study was to evaluate the indicators of nervous system changes occurring in patients with ALMS and BBS using optical coherence tomography (OCT) and magnetic resonance spectroscopy (MRS) methods compared to a group of healthy subjects. The OCT results showed significantly lower macular thickness in the patient group compared to the control group (*p* = 0.002). The MRS study observed differences in metabolite levels between the study and control groups in brain areas such as the cerebellum, thalamus, and white matter. After summing the concentrations from all areas, statistically significant results were obtained for N-acetylaspartate, total N-acetylaspartate, and total creatine. Concentrations of these metabolites were reduced in ALMS/BBS patients by 38% (*p* = 0.0004), 35% (*p* = 0.0008), and 28% (*p* = 0.0005), respectively. Our results may help to understand the pathophysiology of these rare diseases and identify strategies for new therapies.

## 1. Introduction

Primary cilia are non-motile tubule-based structures present in most cells of the human body and responsible for transmitting and combining intracellular and extracellular signals [1]. The products of the *BBS* (BBSsome and chaperonin complex) and *ALMS* genes are responsible for the normal functioning of cilia, and disruption of their function leads to the development of various diseases belonging to the ciliopathy group, including Alström (ALMS) and Bardet–Biedl (BBS) syndromes [2]. The main symptoms characteristic of both diseases, occurring despite the different genes involved [3,4,5,6,7], are early childhood obesity and rod-cone dystrophy leading to blindness [8,9,10]. The molecular basis for these changes is considered to be the incorrect formation and maturation of photoreceptor outer segments [11,12], caused in part by the accumulation of abnormal proteins, including syntaxin-3 [13], due to the loss of proper function of the *BBS* genes expression products. Increased apoptosis in the outer nuclear layer of photoreceptor cells and abnormal localization of rhodopsin, the light receptor in rod photoreceptor cells, to the inner segment have also been observed [14,15].

Given that in both groups of patients with ALMS and BBS syndromes, the causative mutations lead to the coexistence of not only blindness and obesity but also other disorders, such as insulin resistance, type 2 diabetes mellitus (T2DM), bone disorders, kidney failure, and many others that significantly impair quality and length of life, a number of studies are emerging to identify them early and try to find appropriate treatment for this group of patients [16,17,18]. Despite the low prevalence of ALMS and BBS in the general population, attempts are also being made to assess the spectrum of neuronal and ocular changes in this group of patients [19,20,21]. In addition to a detailed ophthalmologic examination, among the generally available examination methods, optical coherence tomography (OCT) appears to be the most accurate method for the detailed evaluation of structural changes in the retinal layers [22]. Moreover, another promising method for understanding cellular mechanisms and their relationship to patient phenotypes is magnetic resonance spectroscopy (MRS). This is a technology derived from nuclear magnetic resonance imaging (MRI) that can detect metabolic changes in vivo, completely non-invasively [23].

The purpose of our study was to evaluate the indicators of nervous system changes present in patients with ALMS and BBS using OCT and MRS methods compared to a group of healthy subjects.

## 2. Materials and Methods

The study protocol was approved by the University Bioethics Committee at the Medical University in Lodz, Poland (RNN/343/17/KE). Patients and/or their parents gave written informed consent for participation in the study. The study protocol adhered to the tenets of the Declaration of Helsinki for research involving human subjects.

Forty eyes of 20 patients with ALMS and BBS were analyzed. The median age of the patients was 17.5 years (8.25–21.5), and the study population included 6 females and 14 males. All patients had a diagnosis of ALMS and BBS based on accepted clinical citation criteria and confirmed by molecular genetic testing, as described earlier [17,24]. Detailed characteristics of the study groups are shown in Table 1. The control group consisted of 30 subjects (60 eyes) with a median age of 12.6 years (9.1–18.8), without obesity or diabetes, age-matched to the study group (*p* = 0.70).

### 2.1. Ophthalmic Examination

The study participants underwent a detailed ophthalmic examination that included the following:Best corrected visual acuity (BCVA) in decimal scale derived from Snellen charts for distant vision (LCD Frey CP-400; Frey Sp.J., Piaseczno, Poland);Intraocular pressure measurement with the Icare ic200 (Icare Finland Oy, Helsinki, Finland);Assessment of the anterior and posterior segment of the eye in a slit lamp (SL-D2; Topcon Inc., Paramus, NJ, USA) using Volk 90 D lens (Volk Optical, Mentor, OH, USA);Corneal topographic and keratometric analyses using Placido disk videokeratography (Keratograf 4; Oculus Inc., Wetzlar, Germany);Optical coherence tomography (OCT; 3-D Maestro II; Topcon, Tokyo, Japan);Topcon 3D-OCT 1000 image generation using a 3D scan corresponding to a 6 mm by 6 mm cube composed of 128 horizontal survey lines of 512 A-scans each;RNFL scanning protocol, which carries out a 360-degree circular scan with a diameter of 3.4 mm centered on the optic disc.

In the study group, tests were performed twice: at the first visit and at the follow-up visit after an average of 3 years (Me 3.05 (2.30–3.47)).

An ophthalmological examination at the first visit was performed on all subjects in the study (n = 20) and control groups (n = 30). In 6 subjects from the study group, because of increased nystagmus, intellectual disability, and the resulting lack of cooperation, the ophthalmic examination was performed only in the basic scope and included a slit-lamp examination of the anterior segment of the eye.

OCT examination at the first visit was feasible in 10 patients in the ALMS/BBS study group, aged 12.9 (7–17.5) years.

Follow-up was carried out in 7 subjects in the study group, and in 6 of them, OCT was performed.

### 2.2. Magnetic Resonance Spectroscopic Analysis

Ten patients with ALMS and BBS who were unable to undergo a complete ophthalmologic examination, including OCT, because of disease progression, including blindness and/or increased nystagmus, underwent spectroscopic analysis on the second visit during a central nervous system (CNS) MRI scan to look for possible exponents of neurodegeneration. The results of the MRS analysis of the study group with BBS and ALMS were compared to those of 17 healthy individuals. Patients were age-matched; the median age of the patients was 20.6 years (15.2–24.3) and 22.4 years (19.9–28.2) in the control group (*p* = 0.25).

All MRI examinations were performed on a 1.5T Ingenia scanner (Philips, USA). First, a T1-weighted 3D TFE sequence was performed to plan the MRS voxels. MRS was performed using a single-voxel technique with TE = 35 ms. Each voxel was shaped like a cube, cuboid, or trapezoid (depending on the anatomical area) with walls with a minimum length of 10 mm and a maximum length of 20 mm; the minimum voxel volume was 1 mL, and the maximum voxel volume was 8 mL.

One voxel was placed in each patient in each of the following locations:-Center of the bridge;-Thalamus;-White matter of the cerebral hemispheric center (white matter);-White matter of the cerebellar hemisphere (cerebellum);-Hippocampus.

For symmetrical brain structures, radiographers were asked to place the voxel on the left side of the brain and cerebellum if possible; in case of anatomical difficulties, voxels were placed on the right side. MRS sequences were performed twice for each patient—with and without water saturation. After the acquisition, MRS data were transferred to Tarquin software version 4.3.10 (free online MRS software), and neural tissue metabolite values were calculated fully automatically.

### 2.3. Statistical Analysis

The normality of the distribution was verified using the Shapiro–Wilk test. The Wilcoxon test was used to compare differences between parameters assessed in the first visit and in the follow-up visit. Comparison of parameter differences between the study groups was performed using a nonparametric Mann–Whitney U test. The OCT measurements were averaged for both eyes. A Spearman correlation test was used for correlation analysis. Categorical variables were presented as numbers with corresponding percentages, and continuous variables were presented as medians with interquartile ranges (IQRs). Results with *p*-values < 0.05 were considered statistically significant. Analyses were performed using Statistica 13.3 PL software (Statsoft, Tulsa, OK, USA).

A nonparametric Mann–Whitney U test was also used to calculate differences in age and the concentrations of metabolites in the brain between ALMS/BBS patients and the control group. In the case of metabolites, obtained *p*-values were corrected by Benjamini–Hochberg false discovery rate (FDR). Statistical analysis was performed using Mass Profiler Professional 15.1 software (Agilent, Santa Clara, CA, USA). GraphPad Prism 9.5.1 software (GraphPad Software, Boston, MA, USA) was used to prepare column plots.

## 3. Results

Visual acuity in the study group of patients ranged from 1.0 (on the decimal scale) to no light perception (NLP). Four people were blind and had no light perception in any eye (20%); one person had NLP in one eye. Only one person had full visual acuity in both eyes (5%).

On ophthalmologic examination, significantly frequent observations were made in the patient group: increased photophobia (in 6 people—30%), high corneal regular hyperopic astigmatism (in 5 people (50%) at values above 5 diopters), horizontal pendular nystagmus (in 13 people—65%), strabismus (in 7 people—35%), abducens nerve palsy leading to impaired eye movement (3 people—15%), ptosis (1 person—5%), cataracts (11 patients, 22 eyes (55%)), retinal pigmentary degeneration (4 people, 8 eyes (28%)), maculopathy (2 people, 4 eyes (14%)), granular appearance of the macula (5 patients; 10 eyes (36%)), optic nerve atrophy (5 people, 10 eyes (35%)), and optic nerve drusen (in 1 patient in both eyes—7%). In one person in one eye (6%), we found the presence of an epiretinal membrane in the macula. In two subjects in the study group (in four eyes (13%)), the intraocular pressure was above 25 mmHg despite maximal topical anti-glaucoma therapy. In one BBS patient, we observed iridodialysis and abnormalities of the peripheral portions of the corneal endothelium forming a membrane toward the iris in one eye (2.5%), resembling iridocorneal endothelial syndrome (ICE).

The percentages were calculated as the ratio of the number of observations to the number of patients actually examined for each symptom.

The results of the ophthalmologic examination in the control group showed only refractive disorders of low severity, while other parameters did not differ from the norm for gender and age.

In the OCT study comparing the two studied groups, there were no statistically significant differences in measurements of optic disc parameters, such as the optic disc area (*p* = 0.714), cup volume (*p* = 0.156), and thickness of the peripapillary retinal nerve fiber layer (*p* = 0.333) in any quadrant of the optic disc. There were also no differences between the study groups in the value of macular volume (*p* = 0.113) and central macular thickness (*p* = 0.571), as expected when comparing eyes with normal outer retina and eyes of patients with retinal dystrophies. A statistically significant difference between the groups was shown in average macular thickness (*p* = 0.002).

A detailed comparison of OCT findings between the study and control groups obtained at the first visit is provided in Table 2.

OCT performed at the second time point in the study group after a mean follow-up of 3 years showed no significant differences from the study at the first time point (Table 3).

On ophthalmological examination at visit 2, which took place 4.6 years after visit 1, we observed a significant decrease in visual acuity from 0.5 to 0.3 on the Snellen chart in both eyes in one BBS patient, patient #16 (Table 1). In the remaining patients from the study group, there were no significant differences in the results of the ophthalmological examination between visits 1 and 2.

In addition, 10 subjects from the study group and 17 controls underwent MRS examinations of the brain, and the concentrations of metabolites in its various areas were determined from the recorded spectra. Statistical analyses of the obtained data were then carried out, comparing metabolite concentrations between the control and study groups in individual brain areas and for the sum of concentrations from all areas. The levels of metabolites between the study and control groups differed in brain areas such as the cerebellum, thalamus, and white matter. For the white matter, there were reduced levels of total choline (TCho) by 27% (*p* = 0.04), total creatine (TCr) by 29% (*p* = 0.05), N-acetylaspartate (NAA) by 53% (*p* = 0.001), and total N-acetylaspartate (TNAA) by 45% (*p* = 0.005). Similar results were obtained for the thalamus. In addition to reduced concentrations of NAA (−53%, *p* = 0.003), TNAA (−45%, *p* = 0.007), and TCr (−45%, *p* = 0.0006), a 53% (*p* = 0.01) reduction (*p* = 0.01) in creatine (Cr) and 40% (*p* = 0.008) in phosphocreatine (PCR) were also observed in the thalamus of patients in the study ALMS/BBS group. Also, in the case of cerebellum, subjects in the study group (compared to controls) had reduced concentrations of NAA (−44%, *p* = 0.02), TNAA (−49%, *p* = 0.0008), and TCr (−38%, *p* = 0.01), as well as lower concentrations (by 46%, *p* = 0.04) of total lipids and macromolecules (TLM13). After summing the concentrations from all areas, statistically significant results were obtained for three metabolites (NAA, TNAA, and TCr). The concentrations of these metabolites were reduced in ALMS/BBS patients compared to the control group by 38% (*p* = 0.0004), 35% (*p* = 0.0008), and 28% (*p* = 0.0005), respectively (Figure 1).

## 4. Discussion

The neuroretina is a functional unit of the central nervous system and, like many other central nervous system structures, contains a huge diversity of neuronal types [25]. Several pathogenic mechanisms, such as anterograde trans-synaptic degeneration following cone-rod dystrophy, primary myelin derangement, and neural occipital reorganization due to chronic visual deafferentation, seem to contribute to the optic pathway derangement in ALMS and BBS [26]. In patients with ALMS, early degeneration of retinal cones may reduce the supply of visual stimuli before the stage of synaptic revision, which occurs during the first 8 months of life, possibly redirecting the maturation of connections [27]. Consequently, rod degeneration may occur at the stage of reduced cortical plasticity, leading to atrophy and decreased connectivity. In patients with ALMS, the timing of retinal degeneration may, therefore, affect the stimulus-driven development of the attuned visual cortex in different ways [28].

Neurological defects are a common finding in many ciliopathies, highlighting a key role for primary cilia in brain development [29]. Primary myelin abnormalities have been reported in the central and peripheral nervous systems of ALMS patients. Diffuse myelin damage has been observed in the brains of ALMS patients. The presence of these abnormalities in the occipital white matter, including optic radiations, has been confirmed. A similar pattern of damage has been described in other diseases characterized primarily by myelin damage, such as relapsing-remitting multiple sclerosis [30].

When analyzing the visual acuity of our patients, even in the first decade of their lives, we observed significantly reduced values, including blindness in both groups of patients. Retinal dystrophy in ALMS tends to be more severe, with an earlier onset and subsequent progressive progression than retinal dystrophy in BBS [7].

In ALMS, nystagmus and photophobia are present from the first months of life, and an abnormal cone electroretinogram develops before 6 months of age. Vingolo et al. [31] suggested a lack of macular development and abnormal persistence of early structural organization of the retina in the course of ALMS, which may explain why these patients experience earlier and more severe vision loss than in typical cone-dystrophies. Legal blindness usually develops in the second decade [7]. Indeed, when analyzing our ophthalmological examination results in the group of ALMS patients, we observed a higher incidence of legal blindness.

Limited data are available on OCT findings in patients with ALMS and BBS [32,33]. OCT findings at two time points and brain neuroimaging results are a strength of our study.

The low visual acuity, increased nystagmus, and adaptive disorders present in patients with ALMS and BBS syndromes prevented OCT from being performed in quite a significant number of patients, which is probably why we could not show statistically significant differences in optic disc parameters. In the posterior slit-lamp physical examination, only in the study group did we observe optic nerve atrophy and retinitis pigmentosa, confirming increased retinal degeneration in the course of ciliopathies. Moreover, the OCT results indicated significantly lower macular thickness in the BBS and ALMS patient group compared to the control group.

The lack of significant progression of retinal and optic nerve changes on OCT at the three-year follow-up of our patients may indicate a relatively slow disease course. A second possible explanation for this result may be the fact that the study group in which OCT was successful consisted of patients with such forms of ciliopathy, in whom blindness and advanced retinal degeneration may never occur. However, this hypothesis requires phenotypic-genotypic analysis on a larger number of patients with ALMS and BBS syndromes. Another factor that influenced the results of the study is the late stage of disease in most of our patients. Also, we should highlight the number of patients that could not be examined via OCT in the second time point, and this limits power and interpretation.

On the other hand, the results of the literature regarding peripapillary RNFL thickness in the course of ciliopathy are inconsistent. Some authors observed a thinning of the RNFL in proportion to the pallor of the optic disc [34] caused by degeneration of retinal ganglion cells, atrophy of the inner retina, and loss of nerve fibers [35]. In contrast, other authors found no difference in RNFL thickness or even its thickening by hypertrophic glial cell proliferation [36].

Furthermore, the reduced levels of N-acetylaspartate and its derivatives found in our study may indicate neuronal degradation/reduction and metabolic damage. NAA is considered an important neuronal marker because it is found exclusively in neurons [37] and plays a key role in myelination and signaling processes between axons and glial cells [38]. Some recent findings suggest that its reduced levels are now considered a hallmark of impaired neuronal viability [39] and demyelination [40]. Moreover, our results were consistent with other studies in which lower NAA levels were observed in neurodegenerative diseases, such as Huntington’s disease, Alzheimer’s disease, Parkinson’s disease, and amyotrophic lateral sclerosis (ALS) [37], but also in patients with diabetic retinopathy in the course of T2DM [41].

However, our study has limitations. Because of the low incidence of both syndromes in the Caucasian population, we were able to include a small group of patients in the study. Moreover, in the group of ALMS/BBS patients we studied, there were disorders that strongly affected the ability to perform the eye examination and its progressive assessment in full. It is noteworthy that fundus examination and OCT, in particular, require special patient involvement and good visual fixation, and the patients’ accompanying nystagmus and behavioral disorders provided additional challenges for the researchers. Because of the need to perform MRI under general anesthesia in younger children, MRS evaluation was performed only in older patients, and therefore, it was not possible to correlate OCT parameters with MRS results.

Nevertheless, our results expand the knowledge of these rare diseases and highlight the value of performing spectroscopic analysis of cerebral N-acetylasparginate derivatives in ALMS and BBS.

## Figures and Tables

**Figure 1 genes-14-01784-f001:**
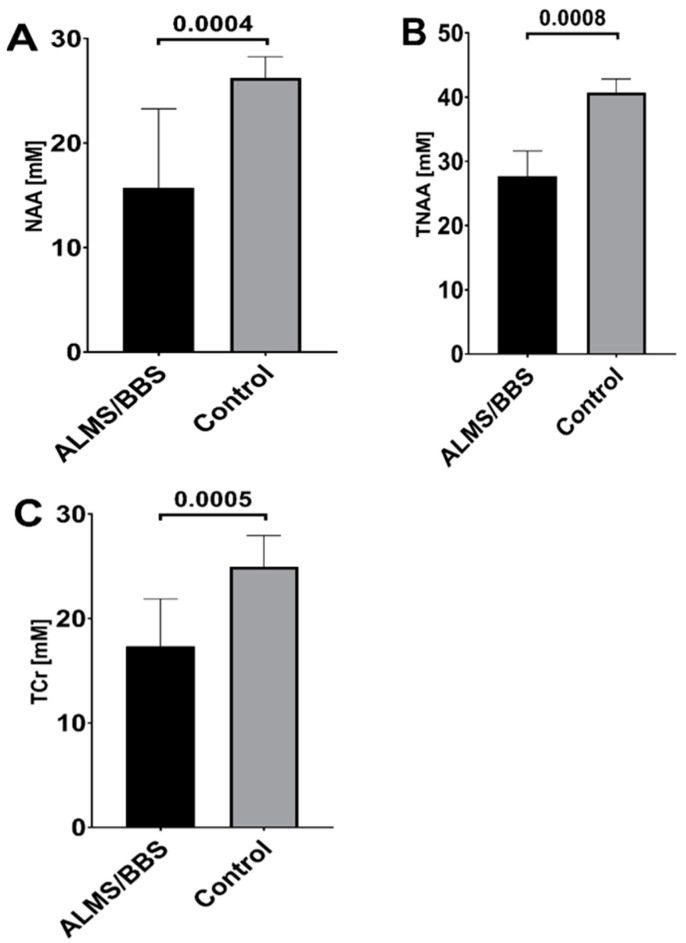
Metabolites with cumulative concentrations in the brain significantly different between ALMS/BBS patients and the control group: N-acetylaspartate (NAA) (**A**), total N-acetylaspartate (TNAA) (**B**), and total creatine (TCr) (**C**). Columns indicate median concentration with IQR.

**Table 1 genes-14-01784-t001:** Detailed characteristics of the study group of ALMS and BBS patients during the first ophthalmological examination.

Patient	Sex	Age (Years)	Diagnosis	Visual Acuity (Snellen Decimal Scale)		Refractive Error (High Hyperopic Astigmatism >5 D)		Photophobia Y/N	Cataract Y/N		Nystagmus Y/N	Retinitis Pigmentosa Y/N	Macula	Optic Nerve Drusen	Optic Disc Pallor/Optic Nerve Atrophy (In Slit-Lamp Examination)	Other Abnormalities in Anterior Segment	IOP >25 mmHg
				R	L	R	L		R	L							
1	M	20	ALMS	LP	LP	-	-	N	N	N	N	-	-	-	-	-	-
2	M	18	ALMS	LP	LP	-	-	N	N	N	N	N	blunted foveal reflex	N	N	N	N
3	M	23	ALMS	NLP	NLP	-	-	Y	Y	Y	Y	N	granular appearance	N	Y	Strabismus	-
4	M	19	ALMS	HM	HM	-	-	N	Y	Y	Y	-	-	-	-	Strabismus	-
5	F	20	ALMS	HM	HM	N	N	N	Y	Y	N	N	blunted foveal reflex	N	N	N	N
6	M	29	ALMS	NLP	NLP	-	-	Y	Y	Y	Y	Y	granular appearance	N	N	Ptosis, Strabismus	N
7	F	25	ALMS	HM	HM	-	-	N	Y	Y	N	-	-	-	-	-	N
8	M	6	ALMS	0.06	0.01	-	-	Y	Y	Y	Y	-	-	-	-	-	Y
9	M	24	ALMS	LP	LP	N	N	N	N	N	Y	-	-	-	-	Limitation of abduction	-
10	F	9	ALMS	NLP	NLP	-	-	Y	Y	Y	Y	N	bull’s eye maculopathy	N	Y	Strabismus	N
11	M	16	ALMS	0.3	0.3	N	N	N	N	N	N	N	N	Y	N	N	N
12	F	8	BBS	NLP	HM	-	-	N	Y	Y	Y	N	granular appearance	N	Y	N	N
13	M	22	BBS	HM	HM	Y	Y	N	Y	Y	Y	N	blunted foveal reflex	N	Y	Strabismus	N
14	M	15	BBS	0.2	0.2	Y	Y	N	N	N	Y	N	N	N	N	Strabismus	N
15	M	17	BBS	1.0	1.0	N	N	N	N	N	N	N	N		N		N
16	F	11	BBS	0.3	0.3	N	Y	Y	N	N	N	Y	ERM	N	N	Iris atrophy and holes in the iris; proliferative and structural abnormalities of the peripheral corneal endothelium	Y
17	M	6	BBS	0.2	0.08	N	N	N	N	N	Y	Y	granular appearance	N	N	Strabismus, Limitation of abduction	N
18	M	6	BBS	LP	LP	N	Y	N	N	N	Y	-	-	-	-	N	-
19	F	18	BBS	0.05	CF	Y	Y	Y	Y	Y	Y	Y	bull’s eye maculopathy	N	N	N	N
20	M	3	BBS	HM	HM	-	-	N	Y	Y	Y	N	granular appearance	N	Y	Limitation of abduction	N

M—male; F—female; R—right; L—left; NLP—no light perception; LP—light perception; HM—hand motion; CF—count fingers; Y—yes; N—no; (-) data not available; ERM—epiretinal membrane.

**Table 2 genes-14-01784-t002:** Comparison of the results of optical coherence tomography of the macula between the study and control groups at the first visit.

	Study Group (n = 10)			Controls (n = 30)			*p*-Value
	Me	IQR25	IQR75	Me	IQR25	IQR75	
Rim area OU (mm^2^)	1.685	1.455	2.270	1.790	1.530	2.015	0.907
Disc area OU (mm^2^)	2.345	1.685	2.575	2.075	1.900	2.560	0.714
Linear CDR OU	0.287	0.087	0.405	0.355	0.150	0.475	0.526
Vertical CDR OU	0.365	0.090	0.420	0.382	0.230	0.485	0.433
Cup volume OU	0.008	0.000	0.044	0.047	0.005	0.110	0.156
RNFL total thickness OU (µm)	113.500	77.000	120.500	102.250	87.000	106.000	0.333
Superior RNFL thickness OU (µm)	113.750	91.500	126.000	128.000	111.000	138.500	0.155
Inferior RNFL thickness OU (µm)	128.500	93.500	144.000	126.750	112.500	142.500	0.987
Temporal RNFL thickness OU (µm)	85.500	65.500	103.000	72.000	64.000	84.000	0.122
Nasal RNFL thickness OU (µm)	79.500	60.500	123.000	71.250	60.500	79.000	0.188
Macular average thickness OU (µm)	217.000	208.000	266.500	279.100	274.000	289.100	0.002 *
Center thickness OU (µm)	194.500	173.000	246.000	209.750	181.500	241.000	0.571
Total macular volume OU (mm^3^)	7.500	6.390	9.600	8.095	7.860	9.750	0.113

Me—median, IQR—interquartile range, CDR—cup/disc ratio, RNFL—retinal nerve fiber layer, OU—oculus uterque, *—indicates significant differences.

**Table 3 genes-14-01784-t003:** Comparison of optical coherence tomography parameters in the study ALMS/BBS group between the two time points.

	Time Point 1. (n = 6)			Time Point 2. (n = 6)			*p*-Value
	Me	IQR25	IQR75	Me	IQR25	IQR75	
Rim area OU (mm^2^)	1.647	1.455	2.120	1.367	1.135	2.360	0.345
Disc area OU (mm^2^)	2.442	1.685	2.575	2.400	2.360	2.590	0.753
Linear CDR OU	0.180	0.105	0.410	0.235	0.000	0.725	0.500
Vertical CDR OU	0.330	0.090	0.420	0.247	0.000	0.750	0.916
Cup volume OU	0.009	0.000	0.044	0.002	0.000	0.200	0.715
RNFL total thickness OU (µm)	114.500	86.500	120.500	90.000	80.000	99.500	0.345
Superior RNFL thickness OU (µm)	117.500	110.500	151.500	77.500	67.000	129.000	0.116
Inferior RNFL thickness OU (µm)	142.000	115.000	150.500	130.750	123.500	134.000	0.916
Temporal RNFL thickness OU (µm)	92.500	83.000	103.000	72.000	70.500	88.000	0.225
Nasal RNFL thickness OU (µm)	76.500	60.500	98.000	78.500	74.000	83.500	0.500
Macular average thickness OU (µm)	209.000	174.000	273.000	204.300	171.100	279.000	0.345
Center thickness OU (µm)	177.000	158.500	231.000	143.500	81.500	210.500	0.249
Total macular volume OU (mm^3^)	7.475	4.715	9.800	5.777	4.835	7.890	0.345

Me—median, IQR—interquartile range, CDR—cup/disc ratio, RNFL—retinal nerve fiber layer, OU—oculus uterque.

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
