# Peer review of "Searching for Effective Methods of Diagnosing Nervous System Lesions in Patients with Alström and Bardet–Biedl Syndromes"

_genes, 2023, doi:10.3390/genes14091784_

Round 1
Reviewer 1 Report
The idea described by the authors in this manuscript is very needed in clinical routine practice. It is very simple to measure macula and fovea thickness for quick diagnostics of pathology. Overall merit of the stated purpose is very valuable. However, there are significant concerns about the goal being achieved based on the description in the manuscript.
Major concerns:
1. Retina thickness is very unspecific parameter: it diminishes in different pathologies. So, the results of statistical significance of retina thinning are obvious without any measurement and do not carry any value because retina is thin in many other diseases, so just measuring retina thickness cannot confirm nor ALMS/BBS neither disease progression rate.
2. Line 230 – no description of OCT findings in the literature – this is not true. Description in the literature usually comes as syndromic form of retinitis pigmentosa which ALMS and BBS are a part of. Retinal abnormalities in BBS are most often consistent with the phenotype seen in retinitis pigmentosa (RP), characterized by rod degeneration that precedes cone degeneration. Certain BBS mutations can also cause isolated, non-syndromic RP or cone-rod degeneration These are some of approx. 50 papers describing what authors consider lacking in the literature:
· Colombo L, Maltese PE, Romano D, Fogagnolo P, Castori M, Marceddu G, Cristofoli F, Percio M, Piteková B, Modarelli AM, Bertelli M, Rossetti L. Spectral-Domain Optical Coherence Tomography Analysis in Syndromic and Nonsyndromic Forms of Retinitis Pigmentosa. Ophthalmic Res. 2022;65(2):180-195. doi: 10.1159/000520329. Epub 2021 Nov 15. PMID: 34781295.
· Sather R 3rd, Ihinger J, Simmons M, Khundkar T, Lobo GP, Montezuma SR. Clinical Characteristics and Genetic Variants of a Large Cohort of Patients with Retinitis Pigmentosa Using Multimodal Imaging and Next Generation Sequencing. Int J Mol Sci. 2023 Jun 30;24(13):10895. doi: 10.3390/ijms241310895. PMID: 37446072; PMCID: PMC10341662.
· Wang Y, Huang L, Sun L, Li S, Zhang Z, Zhang T, Lai Y, Ding X. Ocular findings and genetic test in Alström syndrome in childhood. Exp Eye Res. 2022 Dec;225:109277. doi: 10.1016/j.exer.2022.109277. Epub 2022 Oct 4. PMID: 36206858.
· Lad EM, Duncan JL, Liang W, Maguire MG, Ayala AR, Audo I, Birch DG, Carroll J, Cheetham JK, Durham TA, Fahim AT, Loo J, Deng Z, Mukherjee D, Heon E, Hufnagel RB, Guan B, Iannaccone A, Jaffe GJ, Kay CN, Michaelides M, Pennesi ME, Vincent A, Weng CY, Farsiu S; Foundation Fighting Blindness Consortium Investigator Group. Baseline Microperimetry and OCT in the RUSH2A Study: Structure-Function Association and Correlation With Disease Severity. Am J Ophthalmol. 2022 Dec;244:98-116. doi: 10.1016/j.ajo.2022.08.013. Epub 2022 Aug 22. PMID: 36007554; PMCID: PMC9712171.
3. Line 288 “They can be used to understand their pathological mechanism”. How? Not a word about it. This statement is too strong not being described in more details.
Minor concerns:
1. Table 1 needs to be redone from layout perspective. It’s hard to see and it’s not descriptive in the present form. May be some shortenings will help to increase descriptiveness: Y/N instead of Yes/No, or to make 4-th column a little larger, so ALMS word is not broken into two lines.
2. Figure 1 is not adding value. Just statement that retina in (any) pathology is thinner than normal retina.
3. Line 201 - visit 2. – at which time point? No explanation in the text.
4. Line 217 – cerebellum or cerecellum?
5. Line 228 - median median concentration – double word
6. Figure 2. It’s better to change layout to save space: A and B in horizontal, C downwards from them.
7. Line 300. Number 1 missing in reference 1.
Reviewer 2 Report
Reviewer comments for Waszcykowska et al. Searching for effective methods of diagnosing neuronal degeneration in patients with AlstrÓ§m and Bardet-Biedl syndromes
The authors describe investigation of neurodegeneration in Alstrom and Bardet-Biedl syndrome using OCT and MRS. OCT findings are not a clear marker of neuronal degeneration and this should be clarified in the manuscript. The MRS findings are of interest, demonstrating significant differences in NAA, TNAA, and TCr metabolites in the brain of controls vs BBS/ALMS patients. I have major concerns regarding the authors’ interpretation of the OCT findings. There is no evidence of ‘neuro-degeneration’ found on the OCT data in your study. The reason for the significant finding in average macular thickness is because study patients have photoreceptor loss (characteristic of retinal dystrophies) and thus have thinner macular thickness than patients with normal retinas. For this manuscript to be considered, the presumption that this finding suggests neurodegeneration needs to be removed.
Running title: Suggest do not include ‘syndromes’ at the end as this is already included in the acronyms.
Introduction:
BBS may also be associated with cone-rod dystrophy. PMID: 25982971 PMID: 2346197
Alstrom syndrome is also associated with cone-rod PMID: 17940554
Line 62-63: are you referring to the ‘retinal nerve-fibre layer?’ OCT measurements in retinal degeneration is plagued with inaccuracy and thickness measurements should be interpreted with caution.
Measuring ‘neuro-degeneration’ with OCT may not be reliable. For example, in non-syndromic retinal degenerations (e.g., RHO-mediated retinitis pigmentosa) there will be thinning on OCT but CNS function/cognition/etc is normal thus OCT in isolation is not a reliable marker of neurodegeneration.
Methods:
Table 1. Why are patients not organized by diagnosis? i.e., all BBS together and all ALMS together? For categories marked ‘NA’ was data not available or is this indicating a negative? If data unavailable, mark with ‘-‘ as these data points are all ‘applicable’ to these conditions. These two conditions should both uniformly have ‘retinitis pigmentosa’ – how do patients not have this with the diagnosis? Optic nerve atrophy is typically a secondary feature of retinal dysfunction in RP conditions including BBS. For the purpose of your study, the presence of strabismus is only relevant in the context of being unable to acquire an OCT scan.
What type of OCT was done? Macular cube? RNFL/GCC of peripapillary retina? What measurements were deemed relevant/important?
Which VA format was used? Decimal or LogMAR? In the results, you discuss a range of 1.0 to NPL. Is the 1.0 equivalent to 6/6 (20/20) or 6/60 (20/200)?
With regard to MRS, you mention this is done of the CNS. Do you mean brain or brain and whole spinal cord?
Results
Line 153: specify what full visual acuity means in this context.
Lines 156/157: nystagmus -was this a jerk/motor nystagmus or was this a sensory pendular nystagmus (more typical of retinal dystrophy).
159: RAPD – how was this measured? Typically pupillary reflexes in the presence of retinal dystrophy are sluggish. You also do not state what % of patients had this finding.
162: How is 1 eye 6%? 1 eye of 20 is 5% or does this refer to the proportion of eyes which were able to be imaged with OCT? What was the vision in this ERM eye? ERM typically represents migration of modified RPE cells through the neurosensory retina.
164: suggest change the wording to ‘despite maximal topical anti-glaucoma therapy.’ Again, check your %, 4 eyes of 20 = 20%.
167: Again check % 1 eye of 20 = 5%, not 2.5%
168: What symptoms did the patients describe?
180: Suggest add ‘…as expected when comparing eyes with normal outer retina and eyes of patients with retinal dystrophies.’ Figure 1 is not relevant as this would be true when comparing any control to any group of patients with retinal dystrophy and does not represent a measurement of neuro-degeneration.
Line 191: Suggest remove this paragraph. 1) There is no significant correlation identified here. 2) This represents the effect of retinal dystrophy on visual acuity, a known association, which is not a sign of neurodegeneration.
Line 201: please clarify which patient had this deterioration. Did they have initial maculopathy or RP? Progression would be expected over 3 years in a patient with retinal dystrophy.
Line 205: clarify MRS, not MRI.
Discussion
Line 230: There are a plethora of publications relating to the OCT findings of retinal dystrophies including BBS/ALMS. PMID: 33729075. PMID: 17980398.
247: Be cautious with using retinal dystrophy and neuro-degeneration synonymously. BBS and ALMS are well known syndromic retinal dystrophies. Photoreceptors are highly specialised modified cilia in which the chemistry of the visual cycle is impaired causing progressive damage to photoreceptors. Perhaps define what you mean by ‘neuro-degeneration’ in the intro to clarify what you are claiming by looking at isolated outer retinal changes on OCT.
Lines 249 – 252: Reference 24 describes the use of RNFL measurements in patients with MS (a primary neurological disorder). These patients did not have a retinal dystrophy, which can cause retinal thinning and secondary thinning of RNFL. Additionally, you did not demonstrate statistically significant differences in any of your RNFL measurements between the study and control groups. The average macular thickness (where the only significant difference in OCT data was found) represents the full thickness of the retina, including the photoreceptors which are atrophic/degenerated in the BBS/ALMS group, explaining the difference in measurement.
256: This statement is likely true as loss of photoreceptors (the primary site of retinal disease in BBS/ALMS) precedes downstream loss of inner retina (e.g., RNFL) in RP and other retinal dystrophies.
English grammar is good. Small corrections only.
Round 2
Reviewer 1 Report
The manuscript looks much better after revision. All my comments have been addressed. The paper can be published now.
Author Response
Thank you for your thoughtful and thorough review of our manuscript. We are very grateful and would like to thank you for appreciating our work.
Reviewer 2 Report
The authors have adequately addressed many of the comments. However, the use of OCT to determine neurodegeneration is misleading. Thinning of the inner retina including the RNFL/GCL is anterograde (ie due to loss of photoreceptor function, the primary site of many IRDs) not retrograde, due to CNS disease.
‘An indirect ophthalmoscope was used as the light source. RAPD was observed in 1 ALMS patient in 2 eyes.’ It is not possible to have an RAPD in both eyes of the same patient. RAPD is a sign of asymmetrical retinal or optic nerve disease between the 2 eyes and is a sign of pre-chiasmal dysfunction. Based on your response, I do not believe any of your data on RAPD is valid. Please remove this from the manuscript.
Abstract line 19: remove the comment re: trend toward correlation.
Line 174: why do you think there were no major differences in OCT parameters on follow up? Was there already advanced/end-stage retinal atrophy at the first visit? This also come sup in line 249. With the exception of patient 15, everyone had very poor vision supporting the hypothesis of baseline retinal atrophy (leaving little room for further loss of retinal thickness). Line 252: ‘…blindness may never occur…’ – looking at their visual acuity, blindness has already occurred. Please correct this statement.
Line 221: There is no direct correlation between the MRS and OCT findings. BBS and ALMS are retinal degenerations which cause OCT findings on their own (similar to 300 other genetic types of inherited retinal degeneration). They also cause nervous system lesions as their expressed products function in the CNS. If you remove mention of OCT from your study, it will be much more scientifically robust. Trying to create a link between your OCT and MRS findings weakens the reliability of your work!
Line 272: You state that monitoring macular thickness may be a promising marker of progression, yet you report no progression in OCT. You also state that ‘neuroimaging studies’ were used. This an OCT measurement, which is not neuro-imaging. Again, you must not mislead the reader. Please make clear that you have not demonstrated any correlation between OCT findings and MRS findings.
Line 273: You mention ‘early stages’ of disease, yet the overwhelming majority of patients in your cohort had late stage disease as evidenced by their visual acuity and retinal phenotype.
